# ACAT1/SOAT1 Blockade Suppresses LPS-Mediated Neuroinflammation by Modulating the Fate of Toll-like Receptor 4 in Microglia

**DOI:** 10.3390/ijms24065616

**Published:** 2023-03-15

**Authors:** Haibo Li, Thao N. Huynh, Michael Tran Duong, James G. Gow, Catherine C. Y. Chang, Ta Yuan Chang

**Affiliations:** 1Department of Biochemistry and Cell Biology, Geisel School of Medicine at Dartmouth, Hanover, NH 03755, USA; 2Department of Bioengineering, Perelman School of Medicine, University of Pennsylvania, Philadelphia, PA 19104, USA

**Keywords:** cholesterol, cholesteryl esters, acyl-CoA:cholesterol acyltransferase, sterol O-acyltransferase, ACAT inhibitor, lipid rafts, Toll-like Receptor 4, neuroinflammation, Alzheimer’s disease

## Abstract

Cholesterol is stored as cholesteryl esters by the enzymes acyl-CoA:cholesterol acyltransferases/sterol O:acyltransferases (ACATs/SOATs). ACAT1 blockade (A1B) ameliorates the pro-inflammatory responses of macrophages to lipopolysaccharides (LPS) and cholesterol loading. However, the mediators involved in transmitting the effects of A1B in immune cells is unknown. Microglial *Acat1/Soat1* expression is elevated in many neurodegenerative diseases and in acute neuroinflammation. We evaluated LPS-induced neuroinflammation experiments in control vs. myeloid-specific *Acat1/Soat1* knockout mice. We also evaluated LPS-induced neuroinflammation in microglial N9 cells with and without pre-treatment with K-604, a selective ACAT1 inhibitor. Biochemical and microscopy assays were used to monitor the fate of Toll-Like Receptor 4 (TLR4), the receptor at the plasma membrane and the endosomal membrane that mediates pro-inflammatory signaling cascades. In the hippocampus and cortex, results revealed that *Acat1/Soat1* inactivation in myeloid cell lineage markedly attenuated LPS-induced activation of pro-inflammatory response genes. Studies in microglial N9 cells showed that pre-incubation with K-604 significantly reduced the LPS-induced pro-inflammatory responses. Further studies showed that K-604 decreased the total TLR4 protein content by increasing TLR4 endocytosis, thus enhancing the trafficking of TLR4 to the lysosomes for degradation. We concluded that A1B alters the intracellular fate of TLR4 and suppresses its pro-inflammatory signaling cascade in response to LPS.

## 1. Introduction

Cholesterol is an essential lipid for the growth and maintenance of all mammalian cells. Its metabolites, including oxysterols, neurosteroids, bile acids and steroid hormones, play important physiological functions [1] The brain is the most cholesterol-rich organ in the body. Despite constituting 2.1% of the body’s total weight, it contains 23% of the body’s total cholesterol [2,3]. There is an increasing body of evidence that indicates that cholesterol dyshomeostasis is closely linked with several neurodegenerative diseases, including Alzheimer’s disease (AD) [4,5]

Excess free/unesterified cholesterol can be stored as cholesteryl esters (CEs) through the activity of acyl-CoA:cholesterol acyltransferase (ACAT). There are two distinct ACAT genes, *ACAT1* and *ACAT2,* also called sterol O-acyltransferase 1 and 2 (*SOAT1*, *SOAT2*) in GenBank, encoding two distinct enzymes, ACAT1/SOAT1 [6] and ACAT2/SOAT2 [7,8,9]. Both enzymes use long-chain fatty acyl-CoAs and sterols with 3-beta–OH, including cholesterol and various oxysterols, as their substrates [10]. ACAT1/SOAT1 is ubiquitously expressed in essentially all cells, including cells in peripheral tissues and in the brain. ACAT2/SOAT2 is mainly expressed in intestinal enterocytes and in hepatocytes with low levels of ACAT2/SOAT2 found in various peripheral tissues [11]. Both ACAT1/SOAT1 and ACAT2/SOAT2 are integral membrane proteins located in the endoplasmic reticulum (ER) region, and both are allosterically activated by cholesterol or oxysterols, as reviewed in [12]. CEs are part of lipid droplets; they cannot substitute the functions of cholesterol.

The pathological hallmarks of Alzheimer’s disease include extracellular amyloid plaques, composed of amyloid beta peptides (Aβ, especially Aβ1-42), and neurofibrillary tangles (composed of hyperphosphorylated tau). Late-onset Alzheimer’s disease (LOAD) patients also display an increase in CE levels in the vulnerable regions of brain samples (such as the entorhinal cortex) [13]. Additionally, the relevant brain regions of three different AD mouse models were found to have 3- to 11-fold higher CE content than that of controls [13,14]. These findings suggest that CE content positively correlates with AD risk; however, what causes CEs to be elevated in the AD brains is currently unknown.

Excess free/unesterified cholesterol can be stored as cholesteryl esters (CEs). Normally, CE levels in mouse and human brains are very low. In contrast, in the vulnerable regions of brain samples (such as the entorhinal cortex) from late-onset Alzheimer’s disease (LOAD) patients, CE levels were found to increase by 1.8-fold [13]. In addition, in the relevant brain regions of three different AD mouse models, the CE levels were also found to be 3- to 11-fold higher than those in controls [13,14]. These findings suggest that CE content correlates positively with AD risk. What causes CEs to be elevated in the AD brain is an active area of research. For the biosynthesis of CEs, there are two distinct genes, acyl-CoA:cholesterol acyltransferase 1 (*ACAT1*) and 2 (*ACAT2*) (also called sterol O-acyltransferase 1 and 2 (*SOAT1*, *SOAT2*) in GenBank), encoding two distinct enzymes, ACAT1/SOAT1 [6] and ACAT2/SOAT2 [7,8,9]. Both enzymes use long-chain fatty acyl-CoAs and sterols with 3-beta–OH, including cholesterol and various oxysterols, as their substrates [10]. ACAT1/SOAT1 is ubiquitously expressed in essentially all cells, including cells in peripheral tissues and in the brain. ACAT2/SOAT2 is mainly expressed in intestinal enterocytes and in hepatocytes. Low levels of ACAT2/SOAT2 are also detectable in various peripheral tissues [11]. Both ACAT1/SOAT1 and ACAT2/SOAT2 are integral membrane proteins located in the endoplasmic reticulum (ER) region, and both are allosterically activated by cholesterol or oxysterols, as reviewed in [12]. CEs are part of lipid droplets; they cannot substitute the functions of cholesterol.

Recent evidence from several laboratories has cast new light on ACAT1/SOAT1 as a promising molecular target for the treatment of AD [15,16,17,18,19,20,21,22]. AD pathological hallmarks consist of extracellular amyloid plaques, composed of amyloid beta peptides (Aβ, especially Aβ1-42), and neurofibrillary tangles (composed of hyperphosphorylated tau). Mechanistically, the ACAT1/SOAT1 blockade (A1B) performs the following actions: (1) reduces CE and amyloid pathology [15,17]; (2) increases the content of the neuroprotective oxysterol 24(S)-hydroxycholesterol in the 3xTg AD mouse brain [17] and in the AD patient’s induced pluripotent stem cells (iPSC) derived from human neurons of human AD patients [20]; (3) increases autophagy flux leading to the clearance of Aβ oligomers in microglia [19] and the clearance of misfolded tau in neurons [22]; (4) prevents the inhibitory effects of CEs on hyperphosphorylated tau degradation in early-onset Alzheimer’s disease (EOAD) patient-derived neurons [20]; (5) decreases the protein content of mutant full-length human amyloid precursor protein (hAPP) in the brains of the 3xTg AD model [17] and in AD patients’ iPSC-derived human neurons [20]; (6) clears CEs accumulated in myelin-debris-treated microglia that lack the triggering receptor expressed on myeloid cells-2 (TREM-2), which is a risk factor for LOAD [21]. These results show that in various AD models, A1B offers benefits to suppress amyloid and tau pathologies.

Late-onset Alzheimer’s disease (LOAD) is also accompanied by chronic neuroinflammation. For a review see [23]. To our knowledge, whether A1B affects neuroinflammation in AD models has not been reported. We have previously reported that in a mouse model, A1B reduces inflammatory responses in adipose tissue macrophages and protects against diet-induced obesity by genetic inactivation of *Acat1/Soat1* in the myeloid [24]. In addition, in a different mouse model, myeloid *Acat1/Soat1* knockout (KO) attenuates pro-inflammatory responses in macrophages and protects against atherosclerosis [25]. Macrophages and microglia belong to the same myeloid cell lineage. In the central nervous system (CNS), microglia play a key role in mediating neuroinflammation [26]. Based on our previous work in macrophages, we hypothesize that A1B may also suppress pro-inflammatory responses in microglia. Inflammation may be acute or chronic. At the mechanistic level, acute inflammation and chronic inflammation share many commonalities. It is known that a single, systemic administration of lipopolysaccharides (LPS) produces acute neuroinflammation in adult mice through transcriptional pathways activated by classic immune regulators including TLR4 [27,28]. In the current work, we use LPS-induced acute neuroinflammation as the model to test our hypothesis and report our findings.

## 2. Results

### 2.1. ACAT1/SOAT1 Gene Expression Is Elevated in Neurodegenerative Diseases and in Acute Neuroinflammation

In humans and mice, there are two ACAT/SOAT genes. To document the relative expression of ACAT/SOAT in various cell types in the brain, we first retrieved transcriptomics data from the Barres Laboratory [29] and assessed expressions of SOAT1/Soat1 (ACAT1/Acat1) and SOAT2/Soat2 (ACAT2/Acat2) in the cortex of human and mouse brains under normal conditions (Appendix A). The data indicate that in humans, microglia and macrophage are the main cell types that express ACAT1/SOAT1, while, in mice, astrocytes express more Acat1/Soat1. Oligodendrocytes and neurons also express ACAT1/SOAT1 but at lesser levels. In various cell types examined, the expression of ACAT1/SOAT1 mRNA predominates over that of ACAT2/SOAT2.

In the CNS, chronic inflammation produces cytokines that activate microglia. To determine the expression levels of microglial *ACAT1*/*SOAT1* under neuroinflammation conditions, we obtained transcriptomics data from Friedman and his colleagues [30]. These data described gene expression profiles from purified CD11b^+^ CNS myeloid cells isolated from mouse models for various diseases/conditions (Appendix A). The data indicated that in microglia, *Acat1*/*Soat1* is significantly elevated in many neurodegenerative diseases, including several AD mouse models (in both AβPP and tau models), acute inflammation/infection mouse models, and a mouse model for aging (22 months of age). In humans, additional data presented in Appendix A (the second and third bars in brown) showed that ACAT1/SOAT1 is elevated in the disease-associated region of late-onset Alzheimer’s disease (LOAD). Based on these data, elevated CEs observed in the vulnerable regions of LOAD and in mouse models for amyloidopathy [13,14] can be attributed, at least in part, to elevated *SOAT1*/*Soat1* (*ACAT1*/*Acat1*) expression in microglia located in these regions.

### 2.2. Blocking ACAT1/SOAT1 in Myeloid Attenuated LPS-Mediated Acute Neuroinflammation in the CNS

In mouse models of obesity or atherosclerosis, genetic inactivation of *Acat1*/*Soat1* in the myeloid lineage reduces the inflammatory responses seen in macrophages [25,30]. Additionally, we previously reported that A1B using ACAT1/SOAT1 inhibitors reduces pro-inflammatory responses in LPS-treated RAW264.7 macrophages [25]. Microglia and macrophages belong to the same myeloid lineage. We reason that blocking ACAT1/SOAT1 in myeloid lineage cells may also reduce inflammatory responses in the CNS. To test this hypothesis, we used 2-month-old sex-matched Acat1/Soat1^flox/flox^
*LysM*^Cre^ mice (designated as *Acat1^−M/−M^*) and Acat1/Soat1^flox/flox^ littermates as control. The ACAT1/SOAT1 expression in various brain cell types in Acat1/Soat1^flox/flox^
*LysM*^Cre^ mice including microglia, astrocytes and neurons was previously confirmed by our laboratory [19]. We administered a single intraperitoneal injection of 5 mg/kg LPS to elicit acute neuroinflammation or an equal volume of PBS used as the negative control. One day later, we sacrificed the mice and prepared brain homogenates from both hippocampal and cortex regions and performed q-PCR analyses of various genes. Based on our previous work [25,30], the relevant genes that we chose to analyze included *Il1-β*, *Inos*, *Mcp1*, *Cxcl9*, *Cxcl10*, *Il6*, *Cox2* (representing pro-inflammatory genes), as well as *Arg1*, *Erg2*, *Mrc1* and *Ym1* (representing anti-inflammatory genes).

The results (Figure 1A) show that in the hippocampal region at the basal state in both genders, myeloid A1B did not significantly alter the gene expressions of various pro-inflammatory genes. Injection of LPS highly activated the expressions of multiple pro-inflammatory genes; the fold increases occurred by 20- to 500-fold, in a gene-specific manner. The only exception was *Cox2*; its expression did not respond to LPS. Consistent with our hypothesis, myeloid A1B significantly attenuated the LPS-induced gene expressions of *Il1-β* (by ~70%), *Mcp1* (by ~70%), *Cxcl9* (by ~90%), *Cxcl10* (by ~98%) and *Il6* (by ~98%). In addition, A1B tended to decrease the gene expression of *Mcp1* and *Inos* in males, and *Il1-β, Inos* and *Mcp1* in females, but its effect did not reach statistical significance. Additional results (Figure 1B) show that similar findings occurred in the cortex region: LPS injection elicited large increases in the expressions of various pro-inflammatory genes, and A1B in myeloid lineage significantly attenuated the expressions of these genes. In terms of the expressions of anti-inflammatory genes (Figure 1C,D), at the basal state (without LPS) in both hippocampal and cortex regions, A1B in myeloid did not significantly alter the gene expressions of any of these genes in both genders. LPS injected into *Acat1/Soat1*^flox/flox^ mice did not significantly affect the expressions of these anti-inflammatory genes while LPS injected into the *Acat1/Soat1*^−M/−M^ mice caused significant increases in the gene expressions of both *Arg1* (4-fold in males and 2-fold in females) and *Ym1* (10-fold in males and 5-fold in females) in the hippocampal region (Figure 1C). In the cortex region, LPS injected into the *Acat1/Soat1*^−M/−M^ mice elicited a similar stimulating effect on the *Ym1* gene but not the *Arg1* gene (Figure 1D). These results show that in response to acute inflammation, myeloid A1B attenuated pro-inflammatory responses and increased anti-inflammatory responses in the CNS.

In a similar experiment, we compared the LPS-induced responses of pro-inflammatory and anti-inflammatory genes in male *Acat1/Soat1*^flox/flox^ and *Acat1/Soat1*^flox/flox^
*Syn1^Cre^* (neuronal-cell-specific *Acat1/Soat1*^−/−^) mice to examine the effects of A1B in neurons. The results (Figure 2) show that while LPS injection elicited profound increases in various pro-inflammatory genes in both the hippocampus and cortex—unlike A1B in myeloid—A1B in neurons did not significantly alter the inflammatory gene expression patterns caused by LPS injection. These results are consistent with the notion that in the CNS, LPS-mediated acute inflammation occurs mainly via microglia, at least in male mice.

### 2.3. Blocking ACAT1/SOAT1 with ACAT1/SOAT1-Specific Small Molecule Inhibitor K-604 Attenuated the Pro-Inflammatory Responses of LPS-Treated N9 Microglial Cells through TLR4-Mediated Pathway

We next sought to elucidate the action of A1B in response to acute inflammation using a microglial cell culture. The mouse microglial N9 cell line closely mimics the phenotypes of mouse primary microglia. For instance, in terms of studying the effects of A1B on autophagy and on cellular metabolism, we had previously shown that the results obtained in N9 cells could all be replicated in the primary microglia isolated from mouse brains [19]. Other laboratories have also reported that N9 is a suitable cell line with which to investigate the inflammatory response towards LPS treatment [31,32]. Here, N9 cells were grown in 10% serum and pre-treated for 4 h with DMSO (control group) or with 0.5 µM K-604 [33], then treated with 10, 100 or 1000 ng/mL of LPS for 6 h. After treatment, mRNA was extracted, and qPCR was performed to monitor the expressions of seven pro-inflammatory (A) or three anti-inflammatory (B) genes.

The results (Figure 3A) show that the expressions of five of the seven pro-inflammatory genes (*Il1-β, Inos, Cxcl9, Cxcl10* and *Il6*) dramatically increased (by 10- to over 100-fold) in a gene-specific manner in response to the three doses of LPS tested. The results also show that in all five genes, pre-treating cells with K-604 for 4 h significantly attenuated their increased expressions (by 25 to 80% in a gene-specific manner) in response to LPS. Additional results (Figure 3B) show that the expression of one of the anti-inflammatory genes tested (*Arg1*) is significantly increased in response to LPS (by around 5-fold); pre-treating cells with K-604 did not affect the action of LPS. The expressions of the other two anti-inflammatory genes (*Erg2* and *Mrc1*) decreased following LPS treatment, but the difference did not reach significance; pre-treating cells with A1B did not affect the action of LPS on *Erg2* or on *Mrc1*. The expression of *Ym1* was not calculated due to very low expression. Together, these results show that in N9 cells, the pro-inflammatory responses towards LPS mimic those characterized in the mouse CNS. In addition, treating N9 cells with K-604 attenuated the pro-inflammatory responses elicited by LPS, mimicking the anti-inflammatory actions of A1B by myeloid-specific *Acat1/Soat1* KO in the CNS of mice in vivo. These results support the use of N9 cells as a cell culture model to elucidate the actions of A1B.

In cells of the myeloid lineage, the TLR4 plays a major role in mediating pro-inflammatory cascades [34,35]. Activation of TLR4 by LPS or other TLR4 agonists triggers two signaling cascades [36]: the first involves TLR4 and the adaptor proteins TIRAP and MyD88 at the plasma membrane (PM) [37]. This MyD88-dependent signaling is terminated by endocytosis of TLR4 from the PM. The second TLR4 signaling cascade occurs non-canonically, after TLR4 is endocytosed, and involves other adaptor proteins: TIR-domain-containing adapter-inducing interferon-β (TRIF) and TRIF-related adaptor molecule (TRAM) at the endosomes. After endocytosis, TLR4 can cycle back to the PM or enter the lysosomal compartment followed by degradation [37]. Both signaling events involve activating the IκB kinase in the cytosol, which rapidly phosphorylates IκB and leads to its degradation (by the proteasome) and causes the transcription factor NFκB to move from the cytosol into the nucleus to activate the transcription of various pro-inflammatory genes [38]. To test if A1B inhibits LPS-dependent IκB kinase signaling, we treated N9 cells with or without K-604 for 4 h, then acutely exposed cells to LPS for 30 min and used the cell extracts to moniI tor the phosphorylation status of IκB using Western blot analyses. The results show that LPS treatment rapidly increased the phosphorylation status of IκB and pre-treating cells with K-604 significantly abolished the effect of LPS on the phosphorylation status of IκB and significantly slowed down its degradation (Figure 3C,D). These results indicate that in N9 cells, A1B attenuates LPS-mediated inflammation in part by suppressing the ability of LPS to activate the IκB kinase.

### 2.4. Blocking ACAT1/SOAT1 with ACAT1/SOAT1-Specific Small Molecule Inhibitor K-604 Alters the Fate of TLR4 in N9 Microglial Cells

A1B alters the intracellular distribution of TLR4 and increases endocytosis of TLR4 in N9 cells treated with LPS for 30 min. At the PM, TLR4 undergoes endocytosis and membrane recycling involved with the Golgi, endosomes and PM and TLR4 eventually enters the lysosomal compartment for degradation. The rate of TLR4 endocytosis from the PM affects the LPS-induced pro-inflammatory responses {Reviewed in [36]}. Endocytosis of TLR4 can occur through a clathrin-mediated process [39] or through a caveolae/lipid-raft-mediated process [40]. It is known that after LPS stimulation, a portion of TLR4 is enriched at the caveolae/lipid rafts region of the PM [40]. In mouse embryonic fibroblast cells, A1B causes cholesterol translocation from the ER where ACAT1/SOAT1 resides to various membrane organelles, including the trans-Golgi network (TGN) and various other membrane organelles [41]. In the current cell system, we suspect that pre-treating cells with A1B may affect the intracellular distribution of TLR4, especially upon LPS stimulation. To test these possibilities, we first pre-treated N9 cells with DMSO (control group) or with 0.5 µM K-604 for 4 h, then exposed them with or without LPS at 200 ng/mL for a very short period (30 min). A timeline of the experiment is shown in Figure 4A. We then prepared whole cell homogenates and subjected the homogenates to biochemical fractionation using OptiPrep ultracentrifugation. This procedure produces partial separation of various membrane organelles, primarily based on their differences in buoyant density. Here, we aliquoted individual OptiPrep fractions (1–14) for Western blot analysis [42] to monitor the distribution of TLR4. Lysosomal-associated membrane protein 1 (LAMP1) and plasma membrane calcium ATPase (PMCA) were used as the late endo/lysosome(LE/LYS) and PM markers, respectively. The results showed that TLR4 is mainly distributed in two separate fractions: LE/LYS (fractions 2–3) and PM (fractions 6–8), with the contents in the LE/LYS fraction dominating over those in the PM fraction.

Treating cells with K-604 followed by LPS treatment for 30 min significantly increased the content of TLR in the LE/LYS fractions (Figure 4C, top panel). In addition, treating cells with A1B decreased TLR4 by 50% in the PM, both with and without LPS (Figure 4C, bottom panel). These results show that pre-treatment with A1B causes more TLR4 to accumulate in the LE/LYS fractions, especially under acute LPS treatment.

The results described in Figure 4 suggest that A1B increases the endocytosis rate of TLR4, especially upon acute LPS treatment. To test this hypothesis by using a direct assay, we performed an endocytosis assay in intact N9 cells based on the protocol described in [43]. Cells were pre-treated with or without A1B. Next, they were treated with or without LPS for 30 min, then subject to the endocytosis assay by incubating cells with specific antibodies against TLR4. After endocytosis of the TLR/antibody complex, cell-surface-bound antibodies were removed. Cells were then fixed, permeabilized with 0.3% Triton, stained with fluorescent secondary antibodies and viewed and quantitated as the mean fluorescent area per cell under a confocal microscopy.

The results (Figure 5) show that in cells not treated with LPS, pre-treatment of K-604 increased the amount of internalized TLR4 by around 45%. LPS treatment increased the amount of internalized TLR4 by 81%. Pre-treatment of K-604 followed by LPS further increased the amount of internalized TLR4 by 33%. These results support the interpretation that A1B acts by increasing the endocytosis of TLR4 from the PM to the cell interior, especially in cells acutely treated with LPS.

### 2.5. A1B Decreases TLR4 Protein Content in Microglial N9 Cells Chronically Treated with LPS (for 24–48 h)

The results described in Figure 3, Figure 4 and Figure 5 address the effect of A1B on TLR4 in N9 cells acutely treated with or without LPS for 30 min. Enrichment of TLR4 in the endosomes may lead to more TLR4 entering the lysosomes where it becomes degraded [39]. We tested this possibility by monitoring the effects of A1B on TLR4 in N9 cells treated with or without LPS for a much longer period (24 or 48 h). We first used double immunofluorescence staining in fixed permeabilized cells to monitor TLR4 content and cellular distribution; the PM protein marker N-cadherin was used as the control.

The timeline of this experiment is shown in Figure 6A(i) (for A(ii)–A(iv)) and Figure 6B(i)(B(ii),B(iii)). The results (Figure 6A(ii),(iii)) show that without LPS, A1B does not alter the cellular content of the TLR4 protein. In contrast, in cells treated with LPS for 24 h, A1B caused a significant decrease (by 28%) of the TLR4 fluorescent signals. Additional analyses of the imaging results (Figure 6A(iv)) show that either with or without LPS treatment, A1B does not alter the relative distribution of TLR4 between the internal membrane (IM) and the PM. This result suggests that pre-treating N9 cells with A1B causes a reduction in TLR4 when cells are treated chronically with LPS. To strengthen this finding, using the same conditions used in Figure 6A at different time points, we prepared homogenates from cells to determine the total TLR4 contents using Western blot analysis. Vinculin was used as the loading control. The results (Figure 6B(ii),(iii)) show that in cells with or without LPS treatment for 24 h, A1B by K-604 does not alter the TLR4 protein content. In contrast, in cells treated with LPS for 48 h, A1B caused a significant decrease (by 25%) of the TLR4 protein.

This result combined with the results shown in Figure 4 and Figure 5 suggested that in cells acutely treated with LPS, A1B causes TLR4 at the PM to be endocytosed at a more rapid rate. We speculate that in N9 cells chronically treated with LPS for 24–48 h, A1B causes more TLR4 to be degraded, perhaps as a consequence of more TLR4 entering the late endo/lysosomes. Overall, these results obtained in the cell culture (Figure 3, Figure 4, Figure 5 and Figure 6) support the hypothesis that blocking ACAT1/SOAT1 in microglia suppresses the ability of TLR4 in pro-inflammatory signaling in response to LPS by modulating its intracellular fate.

Activation of TLR4 by LPS triggers two signaling cascades [36]. The first involves TLR4 and the adaptor proteins TIRAP and MyD88 at the PM; this signaling process is terminated by endocytosis of TLR4 from the PM. After TLR4 is endocytosed, the second TLR4 signaling cascade occurs non-canonically at the endosomes and involves other adaptor proteins: TRAM and TRIF. In the first signaling cascade, LPS binds to CD14 or other coreceptors such as CD36 and causes the TLR4/LPS complex to move laterally to be concentrated at certain a cholesterol-rich lipid raft microdomain at the PM to participate in pro-inflammatory signaling and to undergo caveolae-mediated endocytosis [40]. We hypothesize that pre-treating cells with A1B causes subtle change(s) of the cholesterol-rich microdomain at the PM such that the TLR4/LPS complex enriched at the lipid raft domain is endocytosed at a more rapid rate, causing the first signaling event to be terminated more effectively. We also hypothesize that in cells treated with LPS for a longer time (24–48 h), A1B causes more TLR4 to accumulate in the lysosomal compartment, causing it to be degraded, thereby depleting the total TLR4 content. After TLR4 is endocytosed, the second TLR4 signaling cascade occurs non-canonically at the endosomes and involves other adaptor proteins: TRAM and TRIF. A1B may or may not affect clathrin-mediated endocytosis of TLR4 and/or affect the second TLR4 signaling cascade.

## 3. Discussion

ACATs/SOATs are potential targets to treat several human diseases, including atherosclerosis [44], certain forms of cancer early literature reviewed in [12,45] and neurodegenerative diseases that include AD [15] and NPCD [41]. There are two Acats/Soats in mammals [11]. To link *ACATs/SOATs* with neurodegeneration, it is important to document the relative expressions of these two genes in the CNS. In the current work, we retrieved data from the literature to demonstrate that in the CNS: 1. under non-disease conditions, in the human brain and the mouse brain, *ACAT1*/*SOAT1* is the primary ACAT/SOAT isoform (Appendix A); 2. *ACAT1/SOAT1* is expressed in various cell types, with microglia expressing the highest level in both humans and in mice (Appendix A); 3. the expression of *Acat1/Soat1* in microglia are significantly elevated in mouse models for acute inflammation by LPS, for chronic inflammation in various neurodegenerative diseases and for aging; and 4. its expression is also significantly elevated in vulnerable (entorhinal cortex) regions of patients with LOAD (Appendix A). It is known that under these conditions—and in the hippocampus and other vulnerable regions of the LOAD patient brain—the levels of the pro-inflammatory cytokine TNF-α are significantly increased [46,47]. In human monocytes/macrophages, the pro-inflammatory cytokine TNF-α significantly increases *ACAT1/SOAT1* mRNA and ACAT1/SOAT1 protein levels [48]. It is possible that in both acute and chronic inflammatory contexts, increases in TNF-α and other pro-inflammatory cytokines play a key role in upregulating ACAT1/SOAT1 [34,49]. The validity of this hypothesis needs to be tested through future experimentation.

In this present work, we used LPS as a pro-inflammatory stimulus and studied the effect of A1B in the microglia after systematic LPS injection. We chose to study LPS instead of other pro-inflammatory stimuli such as TNF-α to follow up with our previous published works demonstrating A1B reducing inflammation in LPS-treated macrophage [25]. Furthermore, while we cannot eliminate the possibility that the CNS effect that we saw in intraperitoneal LPS-injected *Acat1/Soat1*^flox/flox^*LysM*^Cre^ mice is a secondary response, our N9 cell data demonstrated the effect of A1B in LPS-treated microglia and are in agreement with our mouse data (Figure 1 and Figure 3A,B). Additionally, previous published work by Qin et al. demonstrated LPS induced neuroinflammation through microglia activation in the hippocampus, cortex and substantia nigra in as soon as 3 h using a similar LPS treatment regimen [27]. Together, this further supports the importance and relevance of A1B in the brain, specifically in the microglia.

Current results show that cell-type-specific A1B (by genetic inactivation of *Acat1/Soat1*) in microglia, but not in neurons, significantly attenuated the pro-inflammatory responses induced by LPS in vivo (Figure 1 and Figure 2). To pursue the mechanism(s) of action of A1B, we used the mouse microglial N9 cells and treated the cells with K-604. The results show that in cells acutely treated with LPS (for 30 min), pre-incubating cells with A1B significantly increased the endocytosis of TLR4, the major receptor for mediating the LPS-initiated signaling at the PM (Figure 4 and Figure 5). In cells chronically treated with LPS (for 24–48 h), pre-incubating cells with A1B significantly decreased the total TLR4 protein content, presumably as a consequence of enhanced trafficking of TLR4 to the lysosomes, which is the degradative compartment for many cellular proteins.

Overall, our results show that A1B suppresses LPS-mediated neuroinflammation at least in part by modulating TLR4-mediated signaling. In various immune cells, in addition to TLR4, several other TLRs also play important roles in mediating pro-inflammatory signaling cascade events. Our results cannot rule out the possibility that A1B suppresses LPS-induced pro-inflammatory responses by additional mechanism(s) that are independent of TLR4. For example, our data suggest that anti-inflammatory genes such as *Arg1* and *Ym1* are upregulated in myeloid *Acat1/Soat1* genetic KO mice challenged with LPS, where it is well known that M2 markers are not regulated by the NFκB pathway. The mechanism in which *Acat1/Soat1* deficiency induces these markers’ expression requires further investigation.

Additionally, we had previously reported that blocking ACAT1/SOAT1 in macrophages ameliorates pro-inflammatory responses caused by LPS [24]*,* or by cholesterol loading [25]. It is known that in macrophages, TLR4 plays important roles in mediating various chronic pro-inflammatory signals initiated by oxidized low-density lipoprotein [34]. Based on our current results, A1B may suppress oxidized low-density lipoprotein-mediated inflammation by modulating the fate of TLR4. This possibility also needs to be investigated further.

At present, the mechanism by which A1B can cause an increase in endocytosis of TLR4 is unclear. It has been shown that upon activation by various ligands, monomeric TLR4 (and other TLRs) at the PM as a monomer undergoes lateral diffusion, dimerizes and becomes enriched at a certain cholesterol-rich/sphingolipid-rich microdomain (i.e., lipid rafts/caveolae) at the PM [50]. We have recently reported that in mouse embryonic fibroblast cells, A1B causes redistribution of cholesterol contents in multiple cellular compartments [41]. Thus, A1B could alter the cholesterol content of the TLR4-associated membrane microdomain at the PM and cause this domain to undergo caveolae-mediated endocytoses at a more rapid rate. Other possibilities also exist. For example, in addition to affecting its endocytosis rate, A1B may also affect the rates of recycling of TLR4-containing membranes back to the PM and/or its entry to the lysosomes. These and other possibilities will need to be further investigated.

In the CNS, TLR4 is expressed in microglia, astrocytes, oligodendrocytes and neurons. TLR4 recognizes diverse pathogen-derived ligands including LPS and various tissue damage-related ligands including oligomeric amyloid peptide fragment Aβ1-42, heat-shock proteins (especially HSP60 and HSP70), high-mobility group box 1 (HMGB1), etc. It plays a key role in mediating pro-inflammatory responses caused by various infectious and non-infectious agents and is a LOAD susceptibility gene [51]. It will be interesting to examine if A1B suppresses chronic inflammation that occurs in neurodegenerative diseases including AD and related dementias by modulating the cellular fate of TLR4.

A working hypothesis to explain the A1B actions on suppressing LPS-mediated pro-inflammatory signaling cascades in microglia (Figure 7).

## 4. Materials and Methods

### 4.1. Animals and Acute Neuroinflammation Induction

WT, *Acat1/Soat1*^flox/flox^, *Acat1/Soat1*^flox/flox^*LysM*^Cre^ (myeloid-cell-specific *Acat1/Soat1*^−/−^), *Acat1/Soat1*^flox/flox^*Syn1*^Cre^ (neuronal-cell-specific *Acat1*/*Soat1* knockout) mice were all in the C57BL/6J background. The *Acat1/Soat1*^flox/flox^ and *Acat1/Soat1*^flox/flox^
*LysM*^Cre^ mice were generated by crossing *Acat1*/*Soat1*^flox/flox^ mice with *LyzM*^Cre^ mice (The Jackson Laboratory) as previously described [30]. *Acat1/Soat1*^flox/flox^*Syn1*^Cre^ mice were generated by crossing *Acat1/Soat1*^flox/flox^ mice with *Syn1*^Cre^ mice (The Jackson Laboratory). Littermates produced from relevant mice were used to conduct experiments as described in Figure 1 and Figure 2.

Mice were housed in a specific pathogen-free barrier facility under a regular light–dark cycle and fed standard chow. All mouse protocols were approved by the Dartmouth College Institutional Animal Care and Use Committee and followed NIH guidelines. Acute neuroinflammation was induced by intraperitoneal lipopolysaccharide (LPS) injections of 5 mg/kg body weight as previously described [27]. LPS was obtained from Santa Cruz (sc-3535) and dissolved in sterile PBS as a 5 mg/mL stock solution. After 24 h, mice were perfused with sterile HBSS (Corning 21-022-CV) and sacrificed. The hippocampus and cerebral cortex were isolated and homogenized for RNA extraction as previously described [17].

### 4.2. Cell Culture

Mouse N9 microglial cells were maintained in RPMI-1640 with 10% serum at 37 °C with 5% CO_2_ in a humidified incubator as previously described [19]. For ACAT1/SOAT1 inhibition, ACAT1/SOAT1 inhibitor K-604 was first dissolved in DMSO at 5 mM as stock solution and diluted into the culture medium such that the final concentration was at 0.5 µM as previously described [19].

### 4.3. RNA Isolation and qPCR

The procedure was as described previously [24]. Total RNAs were isolated using a TRIzol reagent (Invitrogen). An amount of 2.5 μg of total RNA as treated with DNase I (New England BioLabs M0303) was used to remove any remaining genomic DNA. A total amount of 0.5 μg DNase-I-treated RNA was reverse transcribed using an iScript cDNA Synthesis Kit (Bio-Rad) to prepare cDNA. qPCR was performed using iTaq Universal SYBR Green Supermix (Bio-Rad). The following cycles were performed: an initial denaturation cycle of 94 °C for 5 min, followed by 40 amplification cycles of 94 °C for 15 s and 60 °C for 1 min. Relative quantification was determined using the ΔΔCT method. The mRNA expression values were normalized with β-actin mRNA levels. The following primers were used:***β-actin*** forward: 5′-CAACGAGCGGTTCCGAT-3′, reverse: 5′-GCCACAGGATTCCATACCCA-3′;***il-1β*** forward: 5′-ACAGAATATCAACCAACAAGTGATATT-3′, reverse: 5′-GATTCTTTCCTTTGAGGCCCA-3′;***inos*** forward: 5′-ACATCGACCCGTCCACAGTAT-3′, reverse: 5′-CAGAGGGGTAGGCTTGTCTC-3′;***mcp1*** forward: 5′-CCCACTCACCTGCTGCTACT-3′, reverse: 5′-TCTGGACCCATTCCTTCTTG-3′;***cxcl9*** forward: 5′-GGAGTTCGAGGAACCCTAGTG-3′, reverse: 5′-GGGATTTGTAGTGGATCGTGC-3′;***cxcl10*** forward: 5′-CCAAGTGCTGCCGTCATTTTC-3′, reverse: 5′-GGCTCGCAGGGATGATTTCAA-3′;***il6*** forward: 5′-TCCAGTTGCCTTCTTGGGAC-3′, reverse: 5′-GTACTCCAGAAGACCAGAGG-3′;***cox2*** forward: 5′-AGCAACCCGGCCAGCAATCT-3′, reverse: 5′-CCTGCTGCCCGACACCTTCA-3′;***arg1*** forward: 5′-CTCCAAGCCAAAGTCCTTAGAG-3′, reverse: 5′-AGGAGCTGTCATTAGGGACATC-3′;***erg2*** forward: 5′-GCAAAGGACCTTGATGGAGC-3′, reverse: 5′-GGCCTAAGTTTTCGGAAGGC-3′;***mrc1*** forward: 5′-CTCTGTTCAGCTATTGGACGC-3′, reverse: 5′-CGGAATTTCTGGGATTCAGCTTC-3′;***ym1*** forward: 5′-CATGAGCAAGACTTGCGTGAC-3′, reverse: 5′-GGTCCAAACTTCCATCCTCCA-3′

### 4.4. OptiPrep^TM^ Subcellular Fractionation

Experiments were performed at 4 °C as previously described [20] Briefly, N9 cells were seeded onto 150 mm plates at a density of 10,000 cells/mL (200,000 cells/plate). After 68 and 72 h, 0.5 µM K-604 (or DMSO as control) and 200 ng/mL LPS (or PBS as control) were added into the plates, respectively. Thirty minutes after adding LPS, cells were scraped off into a 1 mL homogenization buffer containing 0.25 M sucrose, 20 mM Tris buffer (pH 7.4), 1 mM EDTA and a protease inhibitor cocktail (Sigma) and were homogenized using a stainless-steel tissue grinder (Dura-Grind, Wheaton). The post-nuclear supernatant (PNS) was loaded onto the top of a 9 mL continuous 5–20% OptiPrep^TM^ gradient in homogenization buffer and was fractionated by sedimentation velocity ultracentrifugation in a SW41 rotor at 40,000 rpm for 3 h; a total of 14 equal fractions were collected from top to bottom for Western blot analyses.

### 4.5. Whole Cell Protein Isolation and Western Blot Analyses

For whole cell protein isolation, cells were harvested in RIPA buffer a containing protease inhibitor cocktail (Sigma) or phosphatase inhibitor cocktail (Sigma) for phosphorylated proteins. Followed by sonication and centrifugation, the protein concentration of the supernatant was determined by Lowry protein assay. The lysates were run on a 10% SDS-PAGE gel and transferred to a 0.45 μm nitrocellulose membrane for 4 h at 300 mA. After blocking in 5% milk in TBST buffer for 1 h at room temperature, the membranes were incubated with anti-TLR4 (Santa Cruz sc-293072), anti-p-IκB-α (Santa Cruz sc-8404), anti-IκB-α (Santa Cruz sc-1643), anti-Na^+^/K^+^-ATPase α1 (Santa Cruz sc-21712), anti-LAMP1 (Santa Cruz sc-20011) or anti-PMCA (plasma-membrane-type Ca2^+^-ATPases) (Santa Cruz sc-271917), along with anti-vinculin (Novus NB600-1293) antibodies or anti-β-Actin antibody (Santa Cruz sc-47778) as protein loading control.

### 4.6. TLR4 Endocytosis Assay

TLR4 endocytosis was assayed as described previously [43]. In brief, N9 cells were seeded on poly-d-lysine-coated glass cover slides in 12-well plates at a density of 150,000 cells/well and went through different treatments as indicated in the legend in Figure 5. Cells were incubated in RPMI (with 5% goat serum) with 1:200 dilution of specific anti-TLR4 (Santa Cruz sc-293072) antibodies at 4 °C for 1 h, then washed twice in cold RPMI to remove unbound antibodies. Subsequently, cells were re-incubated in RPMI containing 10% calf serum at 37 °C in a cell incubator for 1 h. To remove cell-surface-bound antibodies, the cells were washed twice for several seconds in RPMI adjusted to pH 2.0. Cells were then fixed with 4% paraformaldehyde for 10 min at 4 °C, permeabilized with 0.3% Triton for 20 min at 37 °C, stained with goat anti-mouse secondary antibodies coupled to Alexa Fluor 568 for 1 h at 37 °C. Finally, the coverslips were mounted on microslides with a drop of ProLong Gold antifade reagent with DAPI (Invitrogen) for imaging under a confocal fluorescence microscope.

### 4.7. Immunofluorescence Microscopy

Cells were grown overnight on poly-d-lysine-coated glass cover slides in 12-well plates at a density of 150,000–300,000 cells/well. Cells were washed with PBS and fixed with 4% paraformaldehyde for 10 min at 4 °C, then washed three times with PBS and permeabilized with 0.1% or 0.3% Triton X-100 in PBS for 10–20 min followed by PBS washes three times. Then, the slides were blocked for 1 h at room temperature with 5% goat serum in PBS, incubated overnight at 4 °C with 1:100 anti-TLR4 (Cell Signaling 14358, Invitrogen #PA5-32124), 1:100 anti-LAMP1 (Santa Cruz sc-20011) or 1:200 anti-N-cadherin (Invitrogen MA1-91128) antibodies in blocking buffer, then washed three times with PBS and incubated with AlexaFluor dye-conjugated secondary antibodies for 1 h at room temperature. Afterwards, cells were washed three times with PBS, rinsed with double distilled water, and mounted on glass slides with a drop of ProLong Gold antifade reagent with DAPI (Invitrogen). Confocal Z-stack images were obtained by using a the Andor W1 Spinning Disk Confocal system with a Nikon Eclipse Ti inverted microscope. Image analysis was performed using ImageJ software.

### 4.8. K-604

K-604 was custom synthesized by WuXi AppTec in China. Based on HPLC-MS and NMR profiles, purity was 98% in chemical purity.

### 4.9. LPS

*LPS* was purchased from Millipore Sigma, dissolved in sterile PBS and mixed very well. At the very beginning of our experiments, we searched for the optimum LPS concentration (such as 10, 100 and 1000 ng/mL) to maximize the pro-inflammatory responses, as shown in Figure 3. Later, we found that 200 ng/mL is the optimal LPS concentration in order to maximize the A1B effect. That is why we used 200 ng/mL for our later experiments (Figure 4, Figure 5 and Figure 6).

### 4.10. Statistical Analysis

All statistical analysis was performed using Prism8 software (GraphPad). A two-tailed Student *t*-test was used when two values were compared. For multiple comparisons, a one-way ANOVA with a Tukey post-hoc test was used. Error bars indicate SEM. * *p* < 0.05; ** *p* < 0.01; *** *p* < 0.001; **** *p* < 0.0001.

## Figures and Tables

**Figure 1 ijms-24-05616-f001:**
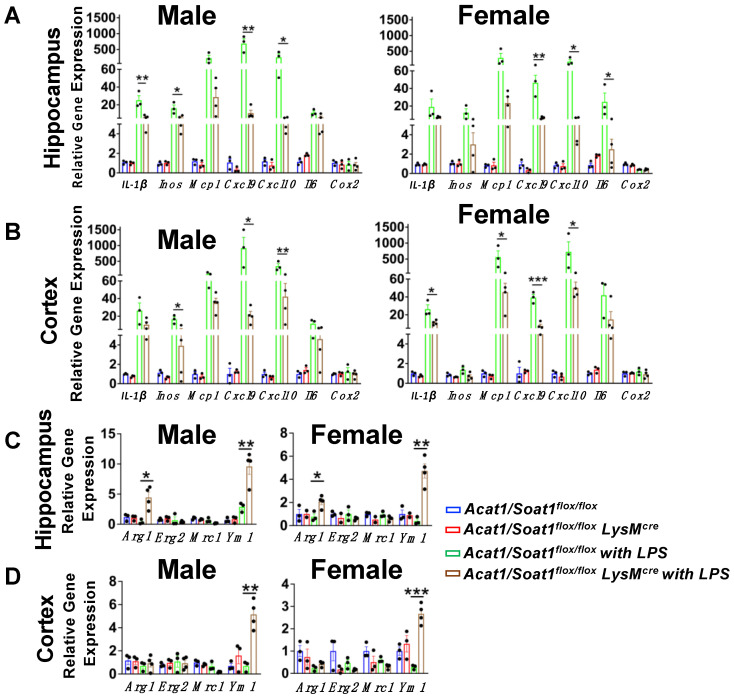
Myeloid *Acat1*/*Soat1* blockade alters the inflammatory gene profile in mouse (**A**,**C**) hippocampus and (**B**,**D**) cortex after LPS injection. Two-month-old *Acat1*/*Soat1^flox^*^/*flox*^ and *Acat1*/*Soat1^flox^*^/*flox*^ LyzMCre (myeloid *Acat1*/*Soat1* knockout) mice as littermates were injected peritoneally with LPS at 5 mg/kg body weight. After 24 h, mice were sacrificed, and mRNA was extracted from hippocampus and cortex. qPCR was performed for pro-inflammatory (**A**,**B**) or anti-inflammatory (**C**,**D**) gene expression. Left panels: male mice. Right panels: female mice. *n* = 4 mice. * *p* < 0.05; ** *p* < 0.01; *** *p* < 0.001.

**Figure 2 ijms-24-05616-f002:**
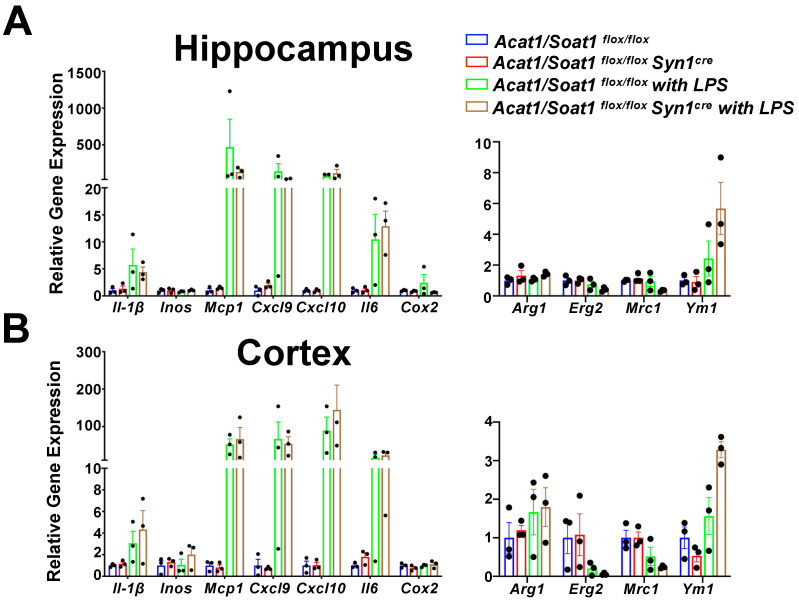
Neuron-specific *Acat1*/*Soat1* blockade did not change the inflammatory gene expression in (**A**) hippocampus and (**B**) cortex from after LPS injection. Two-month-old male *Acat1*/*Soat1^flox^*^/*flox*^ mice and *Acat1*/*Soat1^flox^*^/*flox*^ Syn1Cre(neuronal-cell-specific *Acat1*/*Soat1* knockout) mice as littermates were injected peritoneally with LPS at 5 mg/kg body weight. After 24 h, mice were sacrificed, and mRNA was extracted from hippocampus and cortex. qPCR was performed for pro-inflammatory (left panel) or anti-inflammatory (right panel) gene expression. *n* = 3 mice.

**Figure 3 ijms-24-05616-f003:**
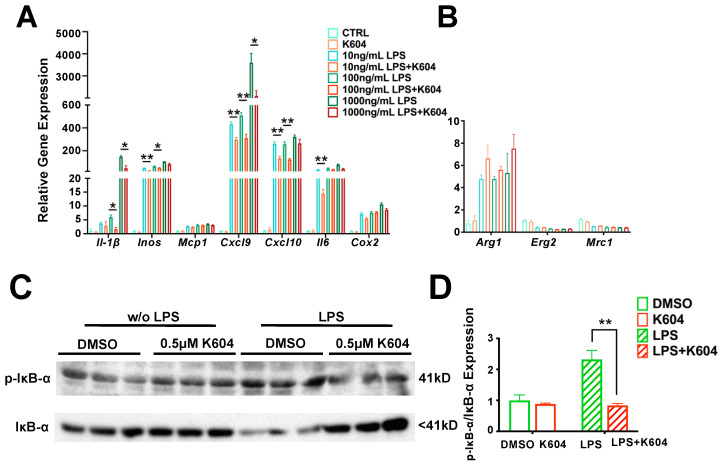
Inhibiting ACAT1/SOAT1 in N9 microglial cell line ameliorates pro-inflammatory responses towards LPS. N9 microglial cells were seeded at 1 × 10^5^ cells per well onto 12-well plates in medium RPMI-1640 with 10% serum. N9 cells were treated with DMSO (control group) or 0.5 µM ACAT1/SOAT1 inhibitor K-604 for 4 h, then treated with 10, 100 or 1000 ng/mL LPS as indicated. After 6 h, mRNAs were extracted, and real-time PCR was performed for (**A**) pro-inflammatory or (**B**) anti-inflammatory gene expressions. (**C**) N9 cells were treated with DMSO (control group) or 0.5 µM K-604 for 4 h, then treated with or without 100 ng/mL LPS for 30 min. Whole cell proteins were used for Western blot to monitor phosphorylation status of IκB-α according to Section 4. (**D**) Western blot quantification for ratio of p-IκB-α/IκB-α. *n* = 3 replicates. * *p* < 0.05; ** *p* < 0.01.

**Figure 4 ijms-24-05616-f004:**
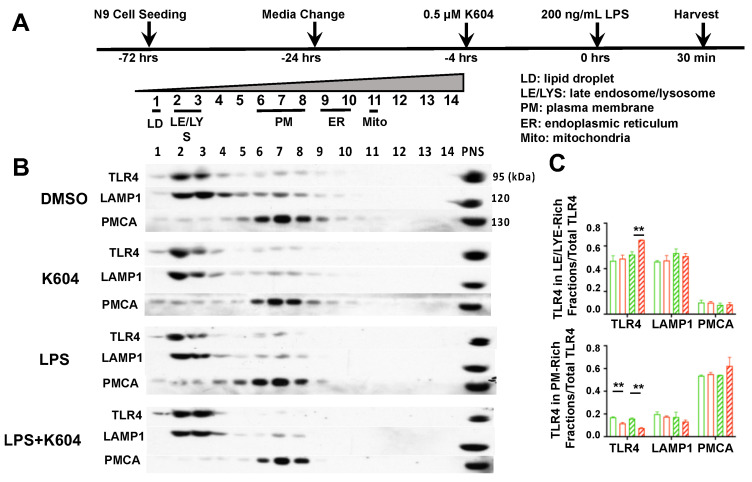
ACAT1 inhibition by K604 causes TLR4 enrichment in late endosome/lysosome in N9 cells acutely treated with LPS. (**A**) Experimental design and organelle marker distribution patterns after OptiPrep^TM^ density gradient ultracentrifugation using the post-nuclear supernatant (PNS) of N9 cell homogenates at far-right lane. Cells were treated with or without LPS from 30 min. (**B**) Fractions were analyzed for TLR4 content as well as for late endosome/lysosome marker LAMP1 and plasma membrane marker PMCA (plasma-membrane-type Ca2+-ATPase). (**C**) TLR4 in LE/LYE-rich fractions and in PM-rich fractions vs. total TLR4 were calculated. *n* = 3 replicates, ** *p* < 0.01.

**Figure 5 ijms-24-05616-f005:**
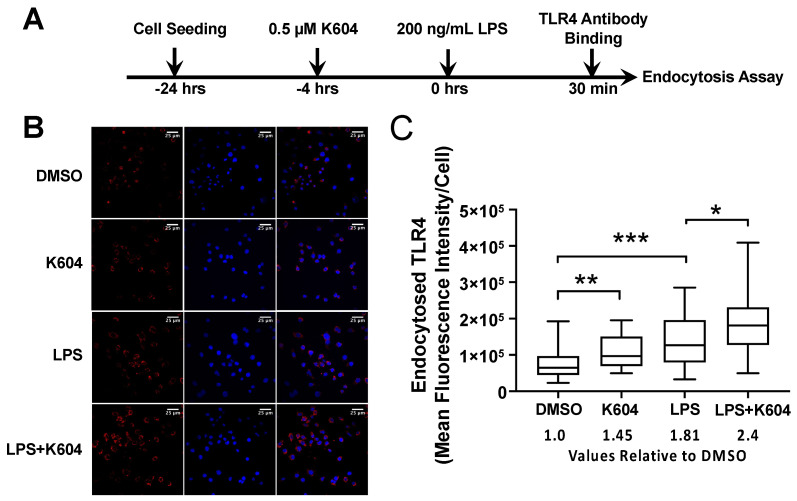
ACAT1/SOAT1 inhibition by K604 increases endocytosis of TLR4 in N9 cells. (**A**) N9 cells seeded on poly-d-lysine-coated glass cover slides in 12-well plates were treated with DMSO or 0.5 µM K-604 for 4 h followed by treatment with or without 200 ng/mL LPS for very short period (30 min). Endocytosis assay was performed as described in Section 4. (**B**) After treating cells with the primary antibodies against TLR4, the secondary antibodies (red) were used to stain TLR in the cell interior; DAPI stain (blue) was used to stain the nuclei. (**C**) Quantification shows the mean fluorescent TLR signal per cell. The TLR signals were normalized by the nucleus signals. *n* = 30 cells/treatment group. * *p* < 0.05; ** *p* < 0.01; *** *p* < 0.001.

**Figure 6 ijms-24-05616-f006:**
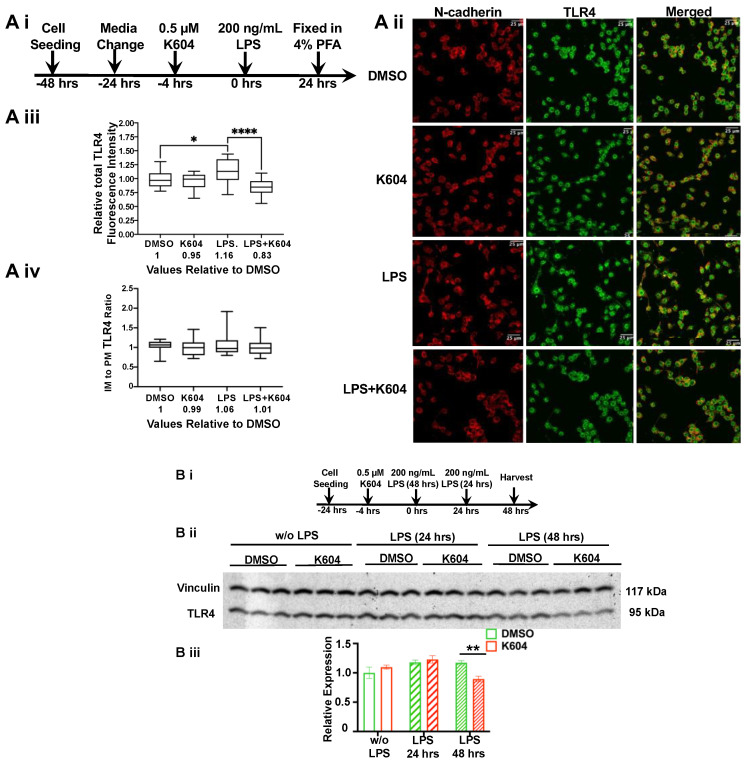
A1B decreases TLR4 protein content in microglia chronically treated with LPS. (**A**) N9 cells seeded at 1 × 10^5^ cells per well on poly-d-lysine-coated glass cover slides in 6-well plates were pre-treated for 4 h with DMSO or with 0.5 µM K-604, then exposed with or without 200 ng/mL LPS for 24 h. Double immunofluorescence staining for TLR4 and for the plasma membrane marker N-cadherin was then performed. (i) Timeline of the experiment. (ii) Representative images demonstrating TLR4 distribution in N9 cells. (iii) Quantification of total TLR4 relative fluorescence intensity per cell, *n* = 15 cells/treatment group. (iv) Quantification of IM/PM TLR4 fluorescence intensity ratio. A total of 15 cells per group were analyzed. The TLR4 signals overlapping with that of N-cadherin are considered as TLR4 at the PM, while those not overlapping are considered as TLR4 at the IM. (**B**) N9 microglial cells were seeded at 2 × 10^5^ cells per well onto 6-well plates in RPMI-1640 with 10% serum. N9 cells were treated with DMSO (control group) or 0.5 µM K-604 for 4 h, then treated with or without 200 ng/mL LPS for 24 and 48 h. At different time points, cells were harvested for protein isolation and TLR4 Western blot analyses. *n* = 3 replicates. Vinculin was used as the protein loading control. (i) Timeline of experiment. (ii) Western blot. (iii) Quantitation of Western blot. * *p* < 0.05; ** *p* < 0.01; **** *p* < 0.0001.

**Figure 7 ijms-24-05616-f007:**
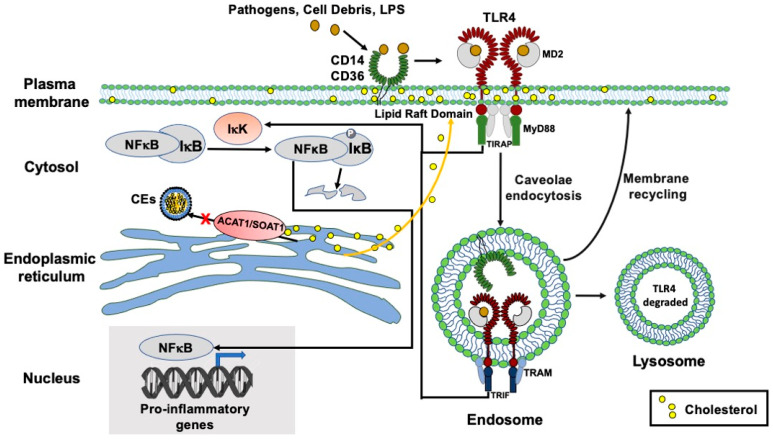
A working model to account for the action of A1B to suppress pro-inflammatory signaling cascade in microglia treated with LPS. At the plasma membrane (PM) of microglia, LPS (or other pathogens/oxidized lipids) binds to the TLR4/MD2 complex (with CD14 or CD36 as coreceptors) to trigger the dimerization of TLR4/MD2 within the lipid raft domain. The adaptor proteins MyD88/TIRAP bound to TLR4/MD2 activate IkK**-**mediated phosphorylation of IκB, causing nuclear translocation of NFκB to activate transcriptions of various pro-inflammatory genes. TLR4/MD2 undergoes caveolae/lipid**-**raft**-**mediated endocytosis. This process terminates its pro-inflammatory signaling activities at the PM. ACAT1/SOAT1 is located at the endoplasmic reticulum. In cells acutely treated with LPS, A1B prevents the cholesterol pool from forming cholesteryl esters (CEs) such that it is translocated to the lipid raft domain at the PM and causes TLR4 to undergo caveolae-mediated endocytosis at a more rapid rate. The TLR4/MD2 complex at the endosome can initiate a second pro-inflammatory signaling event that involves similar but different adapter proteins (TRAM, TRIF, etc.). Whether A1B also affects the signaling activity of TLR4/MD2 at endosomes is currently unknown. The TLR4/MD2 complex at the endosome undergoes membrane recycling to return to the PM; it eventually enters the terminal lysosomal compartment to be degraded. In N9 cells, chronically treated with LPS, A1B may act by increasing the residence time of TLR4 in the lysosomes such that more TLR4 becomes degraded.

## Data Availability

The authors declare that the relevant data are included in the article.

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
