# Peer review of "ACAT1/SOAT1 Blockade Suppresses LPS-Mediated Neuroinflammation by Modulating the Fate of Toll-like Receptor 4 in Microglia"

_ijms, 2023, doi:10.3390/ijms24065616_

Round 1
Reviewer 1 Report
The manuscript is very well written. The research is innovative and relevant, has a clear statement of purpose, and is logically organized.
Author Response
Reviewer #1 Comments and Suggestions for Authors
The manuscript is very well written. The research is innovative and relevant, has a clear statement of purpose, and is logically organized.
We thank the reviewer for his/her positive comments.
Reviewer 2 Report
The work describes the envolment of ACAT1/SOAT1 system in suppressing LPS-induced neuroinflammation in an in vitro model of microglia. In particular, this occurs modulating the intracellular fate of the Tool Like receptor 4 (TRL4).
The work is interesting and relevant for the field: in fact, I think it is can be useful for further progress for modulating neuroinflammation in neurodegenerative diseases, in particular in Alzheimer disease.
However, I have some important questions/considerations the authors would like to answer/comment:
- - The first important consideration is that the number of mice per group is not reported, together with the number of samples considered for OptiPrep, Western blot, TRL4 endocytosis and immunofluorescence analysis.
- - Another important point to clarify concerns OptiPrep images (fig.4B): it is not correct to compare the expression of a protein in samples that have been run in three different gels. To compare them, you should run all the samples all on the same gel and then do the statistical analysis. if this is not possible, then fig.4C should be eliminated.
-- Figures 1 and 2 are too confusing: in particular, the interesting difference in gene expression between male mice and female mice is not appreciated.
- - Probably, the lines 472-513 with fig.7 should be moved in discussion section: in my opinion, it could summarize the results of the work in the final part of the discussion.
TThen, some minor points to clarify:
-- At line 30, please invert “Results revealed that” and “in the hippocampus and cortex”;
-- The sentence at line 32-34 is not clear, probably “ameliorated” should be replace?
-- Reference 6 reported at line 60 is not related to an AD mouse model.
-- Please, control all the abbreviations in the manuscript and add what they stand for the first time used: for ex., ER at line 72, TREM-2 at line 90, at line 166, at lines 180-181, at line 313, at line 629, at line 699.
- - Probably, Figures S1 and S2 should be moved in supplementary data.
- - At lines 160-169, please report which models are considered and re-formulate the sentence, since it is not clear to this reviewer.
- - The sentence at lines 228-231 is not clear. Please, reformulate.
- - Close the bracket al line 289.
- - The authors wrote: “The results also show that, in all 5 genes, pre-treating cells with A1B for 4 hrs significantly attenuated their increased expressions (by 25 to 80%, in a gene-specific manner), in response to LPS (lines 290-292). However, it is not so for all 5 genes and for all three doses of LPS used. applies only to the Cxcl9 gene. Please, correct and clarify.
- - Why was another dose of LPS used for the experiments shown in figure 4 onwards? Why not use one of the doses used in figure 3? Please, explain.
- - In OptiPrep experiment, what were the other cellular fractions considered, in addition to those specified?
- - Why was vinculin chosen as a control instead of actin?
- - Do “Acats/Soats” refer to genes (line 521)?
-- In Materials and Methods sections, please add where K604 was purchased. Please also add for all antibodies used the used dilution.
Author Response
Reviewer #2 Comments and Suggestions for Authors
The manuscript entitled " ACAT1/SOAT1 Blockade Suppresses LPS-Mediated Neuroinflammation by Modulating the Fate of Toll-Like Receptor 4 in Microglia" in which the authors examined the LPS-induced neuroinflammation in microglial N9 cells with or without pre-treatment of ACAT1/SOAT1-specific inhibitor K-604. They found that A1B can alter the intracellular fate of TLR4 and suppress its pro-inflammatory signaling cascade to LPS.
The work is understandable and the topic is important and appropriate for publication in International Journal of Molecular Sciences. The results are interesting. The work is original, and it contains new results that significantly advance the research field. The paper is well written enough to be accepted in its present form. The abstract describes the essential information in the work. The results, supported by figures, are informative. Sufficient references are cited for providing a background to the research. however, the paper suffers from minor concern.
We thank the reviewer for his/her positive comments.
Minor concern:
Please add the number of sample in each figure legend.
We have done so, and added the information at the legends as suggested.
Please write the plasma membrane (PM) not abbreviated in figure 7.
We have done so.
Reviewer 3 Report
The manuscript entitled " ACAT1/SOAT1 Blockade Suppresses LPS-Mediated Neuroinflammation by Modulating the Fate of Toll-Like Receptor 4 in Microglia" in which the authors examined the LPS-induced neuroinflammation in microglial N9 cells with or without pre-treatment of ACAT1/SOAT1-specific inhibitor K-604. They found that A1B can alter the intracellular fate of TLR4 and suppress its pro-inflammatory signaling cascade to LPS.
The work is understandable and the topic is important and appropriate for publication in International Journal of Molecular Sciences. The results are interesting. The work is original, and it contains new results that significantly advance the research field. The paper is well written enough to be accepted in its present form. The abstract describes the essential information in the work. The results, supported by figures, are informative. Sufficient references are cited for providing a background to the research. however, the paper suffers from minor concern.
Minor concern:
· Please add the number of sample in each figure legend.
· Please write the plasma membrane (PM) not abbreviated in figure 7.

Author Response
Reviewer #3 Comments and Suggestions for Authors
The work describes the envolement of ACAT1/SOAT1 system in suppressing LPS-induced neuroinflammation in an in vitro model of microglia. In particular, this occurs modulating the intracellular fate of the Tool Like receptor 4 (TRL4).
The work is interesting and relevant for the field: in fact, I think it is can be useful for further progress for modulating neuroinflammation in neurodegenerative diseases, in particular in Alzheimer disease.
Thank you for your positive comments.
However, I have some important questions/considerations the authors would like to answer/comment:
- The first important consideration is that the number of mice per group is not reported, together with the number of samples considered for OptiPrep, Western blot, TRL4 endocytosis and immunofluorescence analysis.
Thank you, we have added the information in the individual legends.
2 - Another important point to clarify concerns OptiPrep images (fig.4B): it is not correct to compare the expression of a protein in samples that have been run in three different gels. To compare them, you should run all the samples all on the same gel and then do the statistical analysis. if this is not possible, then fig.4C should be eliminated.
Thank you for asking this question. We agree with the reviewer, and have deleted the last figure from Figure 4C: LE/LYE(2+3)/PM(6+7+8) ratio. Now there are only two figures.
- Figures 1 and 2 are too confusing: in particular, the interesting difference in gene expression between male mice and female mice is not appreciated.
Thank you, we have separated the gene expression data between the male and female mice in Figure 1. And we have revised the figures, and rewrote the text in the Result section.
For Figure 2, we only have the male mice data. We have rewrote the text in the Result section.
4 - Probably, the lines 472-513 with fig.7 should be moved in discussion section: in my opinion, it could summarize the results of the work in the final part of the discussion.
We agree with Reviewer 3, we have moved Figure 7 to the Discussion section
Then, some minor points to clarify:
5- At line 30, please invert “Results revealed that” and “in the hippocampus and cortex”;
Thank you, we have changed it.
6- The sentence at line 32-34 is not clear, probably “ameliorated” should be replace?
Thank you, we have changed the word “ameliorated” to “reduced”.
7- Reference 6 reported at line 60 is not related to an AD mouse model.
Thank you, we have corrected it.
8- Please, control all the abbreviations in the manuscript and add what they stand for the first time used: for ex., ER at line 72, TREM-2 at line 90, at line 166, at lines 180-181, at line 313, at line 629, at line 699.
Thank you, we have corrected them.
9 - Probably, Figures S1 and S2 should be moved in supplementary data.
Good suggestions, we have done so.
10 - At lines 160-169, please report which models are considered and re-formulate the sentence, since it is not clear to this reviewer.
We have updated the most recent gene expression data, and edit the text.
11 - The sentence at lines 228-231 is not clear. Please, reformulate.
We have revised the paragraph to:
In a similar experiment, we compared the LPS-induced responses of pro-nflammatory and anti-inflammatory genes in male Acat1/ Soat1flox/flox and Acat1/ Soat1flox/flox Syn1Cre (neuronal cell specific Acat1/Soat1-/-) to examine the effects of A1B in neurons. The results (Figure 2) show that while LPS injection elicited profound increases in various pro-inflammatory genes in both hippocampus and in cortex; unlike A1B in myeloid, A1B in neurons did not significantly alter the inflammatory gene expression patterns caused by LPS injection. These results are consistent with the notion that in the CNS, LPS-mediated acute inflammation occurs mainly via microglia, at least in the male mice.
12 - Close the bracket al line 289.
Thank you, we have done so.
13- The authors wrote: “The results also show that, in all 5 genes, pre-treating cells with A1B for 4 hrs significantly attenuated their increased expressions (by 25 to 80%, in a gene-specific manner), in response to LPS (lines 290-292). However, it is not so for all 5 genes and for all three doses of LPS used. applies only to the Cxcl9 gene. Please, correct and clarify.
We revised the text as:
The results also show pre-treating cells with A1B for 4 hrs significantly attenuated their increased expressions of inos, Mcp1, Cxcl9 and Il6 (by 25 to 80%) in response to LPS at concentration of 10-100 ng/mL.
14 - Why was another dose of LPS used for the experiments shown in figure 4 onwards? Why not use one of the doses used in figure 3? Please, explain.
At the very beginning of our experiments, we searched for the optimum LPS concentration (such as 10, 100 and 1000 ng/mL) to maximize the proinflammatory responses, as shown in Figure 3. Later, we found that 200 ng/mL is the optimal LPS concentration, in order to maximize the A1B effect. That’s why we used 200 ng/mL for our later experiments (Figure 4-6).
We have indicated the above paragraph in the Materials and Methods Section.
15 - In OptiPrep experiment, what were the other cellular fractions considered, in addition to those specified?
We have routinely performed the subcellular fractionation by using OptiPrep in the lab (described in the Materials and Methods section). We have now added the Fractions #7-10 as the endoplasmic reticulum marker, and Fractions #10-11 as the mitochondria marker.
16 - Why was vinculin chosen as a control instead of actin?
Vinculin is a cytoskeletal protein used as a Western blot loading control. It is one of the major components of cell-cell and cell-matrix junctions. Due to its a large size, Vinculins are used as a loading control for high molecular weight proteins.
17 - Do “Acats/Soats” refer to genes (line 521)?
Thank you, we have changed it to Italic.
18- In Materials and Methods sections, please add where K604 was purchased.
K604 was custom synthesis by WuXi AppTec in China. Based on HPLC-MS and NMR profiles, purity was 98%.
Reviewer 4 Report
Comments to authors: The manuscript entitled “ACAT1/SOAT1 Blockade Suppresses LPS-Mediated Neuroinflammation by Modulating the Fate of Toll-Like Receptor 4 in Microglia” seems to be interesting initially, however, the authors just tried to reinvent the same wheel. The hypothesis proposed in this manuscript is vague and lack clear scientific vision. The experimental designs lack novelty, and the writing must be improved at higher degree.
Major Issues:
1. The TLR4/ACAT1 axis in the arena of inflammation is well documented. For instance, the authors begin the manuscript with transcriptomics data from Barres Laboratory and Friedman et al., this seems to be a meta-analysis rather than an original research article, because later the authors correlate more with their earlier publications.
2. The model is confusing and there is no clear rationale provided to use both in vivo and in vitro models in this study.
3. Assessing the gene/protein expression of some inflammatory markers and TLR-4 signalling is minimal compared to the proposed hypothesis or model that the authors detailed in the manuscript.
4. The blot images are speculative, and the quality of the representative images must be updated. For instance, Figure 3C p-IκB-α blot image isn’t convincing to interpret that K-604 reduced the phosphorylation induced by LPS. Likewise, most of the immunoblot images are questionable.
5. TLR4 mobilized into lysosome needs confirmation through specific experiment. Also Figure 6Aii required further enlargement of image to get the clear information objective detail of single cell fluorescence morphology.
6. Western blot images from the original image is not satisfactory.
7. In the whole articles, author self-cited majorly from his previous published article.
8. Cited references are not in sequence.
Minor Issues:
Line 50-52; Check with the reference for the content.
Line 61-62; Reframe the sentence and explain aim and objective of the article.
Line 75-76; Reframe the sentence to "Many evidence from ....." or add recent reference.
Line 104; Sentence is incomplete.
Line 120; Figure S1B, check for the image B which varies from the Barres Lab. once confirm rewrite the result section.
Overall, the manuscript lacks a clear scientific hypothesis and novel experimental designs. The entire study seems to be descriptive and must improve far better before considering for publication.
Author Response
Reviewer #4 Comments and Suggestions for Authors
Major Issues:
- The TLR4/ACAT1 axis in the arena of inflammation is well documented. For instance, the authors begin the manuscript with transcriptomics data from Barres Laboratory and Friedman et al., this seems to be a meta-analysis rather than an original research article, because later the authors correlate more with their earlier publications.
We have revised Figures S1 and S2, and moved these figures as the Supplementary Figures.
- The model is confusing and there is no clear rationale provided to use both in vivo and in vitro models in this study.
Please comments made by Reviewers #1-3.
- Assessing the gene/protein expression of some inflammatory markers and TLR-4 signalling is minimal compared to the proposed hypothesis or model that the authors detailed in the manuscript.
We have revised the sections in the text.
- The blot images are speculative, and the quality of the representative images must be updated. For instance, Figure 3C p-IκB-α blot image isn’t convincing to interpret that K-604 reduced the phosphorylation induced by LPS. Likewise, most of the immunoblot images are questionable.
We used the Image J to quantitate our Western Blots data as indicated.
- TLR4 mobilized into lysosome needs confirmation through specific experiment. Also Figure 6Aii required further enlargement of image to get the clear information objective detail of single cell fluorescence morphology.
We have enlarged the images in Figures 5 and 6.
- Western blot images from the original image is not satisfactory.
We have repeated the Western blots several times, they are reproducible.
- In the whole articles, author self-cited majorly from his previous published article.
We have re-organized the Reference Section by adding more relevant previous published articles.
- Cited references are not in sequence.
Thank you, we have corrected it.
Minor Issues:
Line 50-52; Check with the reference for the content.
Thank you, we have checked all references, and have corrected the mistakes.
Line 61-62; Reframe the sentence and explain aim and objective of the article.
We have corrected it.
Line 75-76; Reframe the sentence to "Many evidence from ....." or add recent reference.
We have changed the sentence to “Recent evidence from several laboratories cast new light on ACAT1/SOAT1 as a promising molecular target for the treatment of AD (Puglielli, Konopka et al. 2001, Hutter-Paier, Huttunen et al. 2004, Bryleva, Rogers et al. 2010, Murphy, Chang et al. 2013, Shibuya, Chang et al. 2014, Shibuya, Niu et al. 2015, van der Kant, Langness et al. 2019, Nugent, Lin et al. 2020).
Line 120; Figure S1B, check for the image B which varies from the Barres Lab. once confirm rewrite the result section.
Thank you for raising this point. This original data was based on an excel spreadsheet of FPKM gene expression data from the Barres lab, downloaded in 2019 at the time of figure generation. Currently the website has download data as "Unavailable feature at this moment." Instead, we visited the Barres lab website and utilized the data that can be visualized by hovering over the error bar. This allowed us to apply original Barres lab data while reformatting it for this figure. Now this revised figure has been included in the Supplementary Figures.
Round 2
Reviewer 4 Report
Author claiming that TLR4 modulate in LPS mediated Neuroinflammation in microglia when pretreated with ACAT1/SOAT1 inhibitory (K604) through endocytosis directed lysosomal degradation. There is no direct supportive data for this. Here N-cadherin, a plasma membrane expression and TLR4 was assessed in immunofluorescence microscopy. But here also need lysosomal mediated degradation of TLR4, yet to clarify.
For the western blot data, the house keeping protein expression band is required.
Author Response
Second Rebuttal letter to Reviewer #4 for manuscript submitted to IJMS ijms-2244702
- Reviewer: Author claiming that TLR4 modulate in LPS mediated Neuroinflammation in microglia when pretreated with ACAT1/SOAT1 inhibitory (K604) through endocytosis directed lysosomal degradation. There is no direct supportive data for this. Here N-cadherin, a plasma membrane expression and TLR4 was assessed in immunofluorescence microscopy. But here also need lysosomal mediated degradation of TLR4, yet to clarify.
Author: Please see Figure 6, Bi, Bii, Biii, and the text in lines #427 to #441, copied below:
“The results (Figures 6A ii, A iii) show that, without LPS, A1B does not alter the cellular content of TLR4 protein. In contrast, in cells treated with LPS for 24-hrs, A1B caused a significant decrease (by 28%) of the TLR4 fluorescent signals. Additional analyses of the imaging result (Figure 6A iv) show that either with or without LPS treatment, A1B does not alter the relative distribution of TLR4 between the internal membrane (IM) and the PM. This result suggests that pre-treating N9 cells with A1B causes reduction of TLR4, when cells are treated chronically with LPS. To strengthen this finding, using the same conditions used in Figure 6A, at different time points, we prepared homogenates from cells to determine the total TLR4 contents by Western blot analysis. Vinculin was used as the loading control. The results (Figure 6B ii, iii) show that in cells with or without LPS treatment for 24-hrs, A1B by K-604 does not alter the TLR4 protein content. In contrast, in cells treated with LPS for 48-hrs, A1B caused a significant decrease (by 25%) of the TLR4 protein.”
We interpret the result described above in lines 443-452, as:
“This result combined with the results shown in Figures 4 and 5 suggest that in cells acutely treated with LPS, A1B causes TLR4 at the PM to be endocytosed at a more rapid rate. We speculate that, in N9 cells chronically treated with LPS for 24-48 hrs, A1B causes more TLR4 to be degraded, perhaps as a consequence of more TLR4 entering the late endo/lysosomes. Overall, these results obtained in cell culture (Figures 3-6) support the hypothesis that blocking ACAT1/SOAT1 in microglia suppresses the ability of TLR4 in pro-inflammatory signaling in response to LPS by modulating its intracellular fate.
- Reviewer: For the western blot data, the house keeping protein expression band is required.
The house keeping protein expression band used here is vinculin (117 ocytosis and by decreasing its total protein content. Overall, these results obtained in cell culture (Figures 3-6) support the hypothesis that blocking ACAT1/SOAT1 in microglia suppresses the ability of TLR4 in pro-inflammatory signaling in response to LPS by modulating its intracellular fate.kDa), please see figure A below. Vinculin (VCL) is a ubiquitously expressed cytoskeletal protein. It is often used as a Western blot loading control (please see figure B below). Because of its large size, vinculins are used as a loading control for high molecular weight proteins.
Figure A (from Proteintech) Figure B (from Labome)
Please see both figures in the PDF documents.
Second rebuttal letter to Reviewer #4 for manuscript submitted to IJMS ijms-2244702
- Reviewer: Author claiming that TLR4 modulate in LPS mediated Neuroinflammation in microglia when pretreated with ACAT1/SOAT1 inhibitory (K604) through endocytosis directed lysosomal degradation. There is no direct supportive data for this. Here N-cadherin, a plasma membrane expression and TLR4 was assessed in immunofluorescence microscopy. But here also need lysosomal mediated degradation of TLR4, yet to clarify.
Author: Please see Figure 6, Bi, Bii, Biii, and the text in lines 427 to 441, copied below:
“The results (Figures 6A ii, A iii) show that, without LPS, A1B does not alter the cellular content of TLR4 protein. In contrast, in cells treated with LPS for 24-hrs, A1B caused a significant decrease (by 28%) of the TLR4 fluorescent signals. Additional analyses of the imaging result (Figure 6A iv) show that either with or without LPS treatment, A1B does not alter the relative distribution of TLR4 between the internal membrane (IM) and the PM. This result suggests that pre-treating N9 cells with A1B causes reduction of TLR4, when cells are treated chronically with LPS. To strengthen this finding, using the same conditions used in Figure 6A, at different time points, we prepared homogenates from cells to determine the total TLR4 contents by Western blot analysis. Vinculin was used as the loading control. The results (Figure 6B ii, iii) show that in cells with or without LPS treatment for 24-hrs, A1B by K-604 does not alter the TLR4 protein content. In contrast, in cells treated with LPS for 48-hrs, A1B caused a significant decrease (by 25%) of the TLR4 protein.”
We interpret the result described above in lines 443-452, as:
“This result combined with the results shown in Figures 4 and 5 suggest that in cells acutely treated with LPS, A1B causes TLR4 at the PM to be endocytosed at a more rapid rate. We speculate that, in N9 cells chronically treated with LPS for 24-48 hrs, A1B causes more TLR4 to be degraded, perhaps as a consequence of more TLR4 entering the late endo/lysosomes. Overall, these results obtained in cell culture (Figures 3-6) support the hypothesis that blocking ACAT1/SOAT1 in microglia suppresses the ability of TLR4 in pro-inflammatory signaling in response to LPS by modulating its intracellular fate.
- Reviewer: For the western blot data, the house keeping protein expression band is required.
The house keeping protein expression band used here is vinculin (117 kDa), please see figure A below. Vinculin (VCL) is a ubiquitously expressed cytoskeletal protein. It is often used as a Western blot loading control (please see figure B below). Because of its large size, vinculins are used as a loading control for high molecular weight proteins.
Figure A (from Proteintech) Figure B (from Labome)

Round 3
Reviewer 4 Report
No comments.